# RNA Activators of Stress Kinase PKR within Human Genes That Control Splicing or Translation Create Novel Targets for Hereditary Diseases

**DOI:** 10.3390/ijms25021323

**Published:** 2024-01-22

**Authors:** Raymond Kaempfer

**Affiliations:** Department of Biochemistry and Molecular Biology, Institute of Medical Research Israel-Canada, Faculty of Medicine, The Hebrew University, Jerusalem 9112102, Israel; kaempfer@hebrew.edu

**Keywords:** PKR activation, RNA splicing, translation, inflammatory cytokine genes, *IFNG* RNA, *TNF* RNA, *globin* RNA, eIF2α phosphorylation

## Abstract

Specific sequences within RNA encoded by human genes essential for survival possess the ability to activate the RNA-dependent stress kinase PKR, resulting in phosphorylation of its substrate, eukaryotic translation initiation factor-2α (eIF2α), either to curb their mRNA translation or to enhance mRNA splicing. Thus, *interferon-γ* (*IFNG*) mRNA activates PKR through a 5′-terminal 203-nucleotide pseudoknot structure, thereby strongly downregulating its own translation and preventing a harmful hyper-inflammatory response. *Tumor necrosis factor-α* (*TNF*) pre-mRNA encodes within the 3′-untranslated region (3′-UTR) a 104-nucleotide RNA pseudoknot that activates PKR to enhance its splicing by an order of magnitude while leaving mRNA translation intact, thereby promoting effective TNF protein expression. Adult and fetal *globin* genes encode pre-mRNA structures that strongly activate PKR, leading to eIF2α phosphorylation that greatly enhances spliceosome assembly and splicing, yet also structures that silence PKR activation upon splicing to allow for unabated *globin* mRNA translation essential for life. Regulatory circuits resulting in each case from PKR activation were reviewed previously. Here, we analyze mutations within these genes created to delineate the RNA structures that activate PKR and to deconvolute their folding. Given the critical role of intragenic RNA activators of PKR in gene regulation, such mutations reveal novel potential RNA targets for human disease.

## 1. Introduction

The RNA-activated protein kinase R (PKR) is an intracellular sensor of stress that plays a central role in antiviral defense and innate immunity [1]. Double-helical RNA activates PKR, which leads to inhibition of protein synthesis through PKR-mediated phosphorylation of the α-chain of eukaryotic translation initiation factor 2 (eIF2α) on Serine 51; transient phosphorylation of eIF2α blocks GTP/GDP exchange essential for the recycling of eIF2 between rounds of translation initiation [2]. Phosphorylation of eIF2α is critical for mounting the integrated cellular stress response [3,4]. Double-stranded RNA produced during viral replication activates PKR, which phosphorylates eIF2α to inhibit overall translation within the cell, resulting in apoptosis of infected cells and thereby preventing virus spread [5]. Interferons induce higher intracellular levels of PKR, rendering PKR an essential mediator of interferon-induced antiviral responses [5].

Effective engagement of the tandem RNA-binding domains in a PKR monomer requires at least 16–18 base pairs of double-helical RNA [6] and *trans*-autophosphorylation of the PKR homodimer that is needed for subsequent kinase activation [7,8] requires a minimum of 33 base pairs of double-helical RNA [6,9].

Both *trans*-autophosphorylation of PKR essential for PKR activation and the consequent phosphorylation of eIF2α are local, transient events within the cell that take place in close proximity to the activating RNA molecule, and are followed promptly by dephosphorylation [10,11,12]. Upon dephosphorylation, PKR becomes inactive again, whereas eIF2α returns to its active state and can fulfill its function in the initiation of translation. Within the broader context of the cell, levels of activated PKR and of phosphorylated eIF2α remain constant. This accounts for the strictly *cis*-acting nature of intragenic RNA activators of PKR that will be analyzed below.

The finding that short RNA sequences within unspliced precursor transcripts (pre-mRNA) or within mature mRNA of human genes possess the exceptional ability to activate PKR, first discovered for the human *tumor necrosis factor-α* (*TNF*) gene [13,14], shortly thereafter for the *interferon-γ* (*IFNG*) gene [15,16], and more recently for the adult and fetal *globin* genes [17], reveals a novel susceptibility for RNA-mediated disease resulting from mutations within these RNA activators of PKR that affect their indispensable function. Indeed, regulation of precursor transcript RNA splicing or mRNA translation via intragenic RNA activators of PKR, illustrated by these examples, is likely to be more widespread within the human genome. Here, we focus on mutations within these inflammatory cytokine and hemoglobin genes that were created to define essential features of the RNA structures that activate PKR and to deconvolute their folding, as first examples of novel targets for RNA-mediated human diseases.

## 2. Results and Discussion

### 2.1. The RNA Activator of PKR within Human IFNG mRNA That Controls mRNA Translation

Before entering into the analysis of mutations within the *IFNG* gene that may lead to RNA-mediated human disease, it is important to consider first the *IFNG* regulatory circuit involving PKR activation. *IFNG* provides the first example of a gene that controls the translation of its own mRNA by over an order of magnitude through the device of local activation of PKR by *IFNG* mRNA [15]. Indeed, overexpression of IFN-γ protein occurs in many autoimmune diseases [18] as well as in toxic shock induced by bacterial superantigen toxins [19,20]. To avoid a harmful hyper-inflammatory response, the *IFNG* gene expresses within its mRNA a 203-nucleotide RNA pseudoknot element located within the 5′-untranslated region (UTR) and the start of the open reading frame (Figure 1) that potently activates PKR locally in *cis* [15,16]. This in turn leads to a local phosphorylation of eIF2α within the vicinity of *IFNG* mRNA that strongly diminishes the rate of initiation of translation of this mRNA while leaving the translation of other mRNA species in the cell intact. Dynamic refolding of the RNA pseudoknot enables *IFNG* mRNA to function both as a potent activator of PKR and as an RNA template for translation [16]. This autoregulatory circuit renders the translation of *IFNG* mRNA sensitive to viral inhibitors of PKR such as Vaccinia virus E3L, leading to significantly higher expression of IFN-γ upon viral infection [15]. PKR is expressed constitutively in the cell [21] to levels that support this *IFNG* autoregulatory mechanism [15].

The *a4* mutation (U9G A14C A29G A45U; yellow in Figure 1) severely impairs the activation of PKR. Restoration of A14 in the *a4* mutant sufficed to restore the ability to activate PKR. The *d1* deletion of the 5′-terminal nucleotides through U9 (cyan, light green, and yellow in Figure 1) leads to nearly complete loss of activation of PKR and increases the translation efficiency of wild type *IFNG* mRNA by more than tenfold. Mutations U9 and A14 each affect the stability of the pseudoknot stem (red text in Figure 1). In PKR knockout cells, translation efficiency of wild type (*wt*) *IFNG* mRNA was increased to that of *a4* and *d1* mRNA. The *a4r* mutation (U64G A69C; yellow in Figure 1) likewise enhanced translation efficiency of *IFNG* mRNA, by 7-fold, whereas the compensatory *a4/a4r* mutation restored the stability of the pseudoknot stem and the activation of PKR and lowered the translation efficiency to almost that of *wt* mRNA. These results render mutation of either nucleotide U9, A14, U64, or A69 within the 5′-UTR of the *IFNG* gene critical targets for inflammatory disease, leading to excessive expression of IFN-γ protein by around an order of magnitude [15].

Moreover, the U7A mutation (light green in Figure 1) abrogates PKR activation and enhances mRNA translation by 6-fold, showing that U7 is critical for proper function [16].

At the junction of helices *S1* and *S2*, the A120:U15 base pair (red in Figure 1) is critical for PKR activation. Mutation of this pair to G:C resulted in near-total loss of the ability to activate PKR. Moreover, mere inversion of the A120:U15 pair sufficed to abolish PKR activation. By contrast, when all of the nucleotides in helix *S1* (U15-A29:U103-A120) except the A120:U15 pair were replaced by a theophyllin aptamer and slippery helix [22], the ability to activate PKR remained intact [16]. These results revealed that the orientation of the A120:U15 pair located at the helical junction in *S1* is critical for PKR activation.

Helix *S2* is likewise critical for PKR activation. Replacement of nucleotides 121–127 (green in Figure 1), including the AUG start codon, by the nucleotides in the opposite strand abolished the ability to activate PKR (mutation *s2a*). By contrast, replacement of nucleotides 163–169 by the nucleotides in the opposite strand left the ability to activate PKR intact (mutation *s2b*). Restoration of base pairing by an inverted S2 helix (*s2ab*) failed to restore the ability to activate PKR. However, replacement of the helical junction base pair U121:A169 within *s2ab* by the original A121:U169 (light green in Figure 1) led to full recovery of the ability to activate PKR, demonstrating that A121-U169 is a critical, orientation-sensitive base pair for PKR activation. Replacement of the bifurcation loop adjoining the AUG start codon (nucleotides 128–162) by pentaloop UUCGU left the ability to activate PKR largely intact, showing that the bifurcation loop is not critical for this property. The A121-G127 strand in helix *S2* can pair alternatively with the U7-C1 5′-terminal strand for PKR activation. Like the A121:U169 base pair, the A121:U7 pair is orientation-sensitive, as seen from the U7A mutation reported above, which abrogates PKR activation. Even when *S2* is intact, deletion of the first six nucleotides C1-U6 (cyan in Figure 1) results in marked loss of the ability to activate PKR, and to complete loss when U7 is also deleted [16].

A single nucleotide change within helix *S3*, A202U (light blue in Figure 1), sufficed to abolish PKR activation. By contrast, the U171A single mutation did retain partial ability to activate PKR. However, U171A could not rescue A202U, as the U171A:A202U double mutation completely lost the ability to activate PKR. This revealed that as for A120:U15, the U171:A202 base pair is orientation-sensitive for PKR activation. Hence, A120 and A202 are potential mutation targets for inflammatory disease. The remainder of helix *S3* is also sensitive to mutation; *s3a* (orange in Figure 1), in which U171-C184 are replaced by the nucleotides in the opposite strand, abrogated the ability to activate PKR [16].

Moreover, U170 adjoining the *S2* helix is located in a sensitive position; mutation U170A (brown in Figure 1) resulted in loss of PKR activation [16].

Formation of a kink-turn (K-turn) is known to depend on purine–purine pairing; a G•A/A•G quartet underlies most K-turns in prokaryotic RNA [23]. Indeed, in the *IFNG* K-turn, mutation G38C (purple, Figure 1) severely reduced the activation of PKR and led to an enhancement of translation efficiency even greater than that caused by the *d1* mutation [15,16]. Moreover, mutation A95G (purple in Figure 1) similarly impaired the ability to activate PKR, pointing to a G•A/A•A motif as essential for PKR activation by *IFNG* mRNA. Typical K-turns have G:C base pairs bordering the bulge [23], yet neither U92C nor G36C mutation (light purple, Figure 1) affected the activation of PKR [16].

Within the loop bordering the pseudoknot stem U64-G70, no detrimental mutations could be detected. However, replacement of the four nucleotides 78CAAG81 (grey in Figure 1) in the short stem by those of the opposite strand (78GUUU81) resulted in near-total loss of the ability to activate PKR, showing that this short stem is critical for function [16].

Thus, throughout its 203-nucleotide sequence, the *IFNG* RNA element that activates PKR is extremely sensitive to single nucleotide mutations, rendering it a prime target for human inflammatory disease.

### 2.2. The RNA Activator of PKR within Human TNF Pre-mRNA That Controls mRNA Splicing

*TNFα* (*TNF*) provides the first example of a gene that regulates splicing of its own pre-mRNA by over an order of magnitude through the device of local activation of PKR within the nucleus by its pre-mRNA, leading to the phosphorylation of eIF2α that enables efficient splicing [13,14,24]. The 104-nucleotide RNA activator of PKR within the human *TNF* gene was designated as the 2-aminopurine response element (2-APRE) [13] because 2-aminopurine, a small-molecule blocker of PKR activation and action, inhibits the splicing of *TNF* pre-mRNA [13,24]. The TNFα cytokine is produced early during an immune response. Upon immune stimulation, *TNFα* mRNA is expressed rapidly, reaching its maximum within 3 h [24], whereas the highly homologous *TNFβ* (*lymphotoxin*) mRNA is still increasing by 24 h. The difference is vested in the *TNFα* 2-APRE, because insertion of this element into the *TNFβ* 3′-UTR sufficed to increase *TNFβ* RNA splicing efficiency by an order of magnitude [13,14]. Increased expression of PKR within the cell causes a strong increase in splicing of *TNF* mRNA [13]. The 2-APRE RNA sequence of 104 nucleotides folds into a compact pseudoknot (Figure 2). This pseudoknot constrains the RNA into two double-helical stacks with parallel axes (Figure 2B) that are each of sufficient length to bind a PKR monomer, enabling PKR dimerization that is needed for its local activation [14]. Within the nucleus, phosphorylation of eIF2α substrate upon PKR activation is not only strictly needed for splicing of *TNF* pre-mRNA but is even sufficient to promote highly efficient splicing [14]. eIF2α phosphorylation upregulates *TNF* mRNA splicing in human peripheral blood mononuclear cells, showing its physiological relevance. Once exported from the nucleus, *TNF* mRNA is translated without inhibition by PKR activation [13,14], presumably because cytoplasmic proteins shield the 2-APRE [13].

Long stem–loop *S1* forms an integral part of the *TNF* RNA activator of PKR (cyan in Figure 2). Helix *S1* forms a domain that is sensitive to mutation. Mutating 19CCAG22 to the four nucleotides in the complementary strand 19GGUC22, yielding mutation *S1a* (blue in Figure 2A), reduced the ability to activate PKR by more than twofold. Mutating in addition 29CUGG32 to 29GACC32 to create double mutation *S1ab* (blue and green in Figure 2A) only slightly improved PKR activation. The loop of helix *S1* in the 2-APRE is also sensitive to mutation. Thus, mutating nucleotides 25CUC27 to 25GCG27, yielding mutation *S1L* (purple in Figure 2A), reduced the ability to activate PKR to the same extent as mutation *S1a*.

The *TNF* 2-APRE pseudoknot is created around two pseudoknot stems, *P1* (red in Figure 2) and *P2* (green in Figure 2). Pseudoknot stem *P1* (red box in Figure 2A) is critical for PKR activation. Mutation of 39CAGC42 to the nucleotides in the opposite strand, 39GUCG42 (*P1a* mutation, light blue in Figure 2A), severely impaired PKR activation, as did mutation of 100GCUG103 to 100CGAC103 (*P1b* mutation, yellow in Figure 2A). Activation of PKR was restored in part by the *P1ab* double mutation.

Pseudoknot stem *P2* is required for full PKR activation. Mutation of 4UUC6 to the nucleotides in the opposite strand, 4AAG6 (*P2b* mutation, orange in Figure 2A), somewhat impaired PKR activation. Activation of PKR was restored largely by also mutating 96GAA98 to 96CUU98 (*P2a* mutation, light purple in Figure 2A), creating the *P2ab* double mutation.

The indispensable role of helix *S3* (orange in Figure 2) bordering *P2* is highlighted by the *S3bP2b* double mutation which changes 93CCAGAA98 into 93GGUCUU98 (pale orange and light purple in Figure 2A), to thereby abrogate the ability to activate PKR.

The *TNF* RNA activator of PKR relies tightly on the base pairs adjoining pseudoknot stem *P1* and at the terminus of stem *P2* (three purple ellipses in Figure 2A). Thus, mutations G13C and G13A each weakened the ability to activate PKR, whereas mutation of U37C largely abrogated PKR activation. This shows that the 13G•U37 wobble base pair (light brown in Figure 2A) at the start of helix *S1* is highly specific for function; neither 13C•U37 nor 13G•C37 can replace it. Each of the A3U and U99A mutations strongly impaired PKR activation, which could not be restored in the A3U U99A double mutation, showing that the 3A:U99 base pair is orientation-sensitive. By contrast, each of the G96C and C6G mutations strongly impaired PKR activation, which could be restored largely by the G96C C6G double mutation.

The long stem–loop *S2* (dark blue in Figure 2B) forms an integral part of the *TNF* RNA activator of PKR. The U48•A78 base pair at the base of long helix *S2* in the 2-APRE (red in Figure 2A) is critical for the ability to activate PKR. Mutations U48A and A78U each severely reduce the activation of PKR by 2-APRE RNA, as does double mutation U48A A78U, showing that the U48•A78 pair is orientation-sensitive. By contrast, mutation of 49CCCUG53 in helix *S2* to 49GGGAA53 (S2aUA mutation) left activation of PKR intact. By contrast, *S2bUA* mutation of 73CAGGG77 in helix *S2* to 73UUCCC77 (light blue in Figure 2A) impaired PKR activation. However, double mutation *S2abUA* impaired PKR activation only weakly, showing that this part of helix *S2* adjacent to the U48•A78 pair is more tolerant to mutation.

The importance of stem–loop *S3* nucleotides (pale orange and grey in Figure 2A; orange in Figure 2B), which joins the two parallel helix motifs *S1-P1-P2* and *S2* within the folded pseudoknot structure (Figure 2B), is documented by the deletion of nucleotides 84–95 (grey and orange in Figure 2A), which weakens the ability to activate PKR.

The six-nucleotide linker (*lin*) (grey in Figure 2), which connects pseudoknot stem *P2* to helix *S1*, is highly sensitive to deletion, as might be expected. Indeed, deleting nucleotides 7AAAC10 (light green in Figure 2A) totally abolishes the ability to activate PKR. Moreover, mutating 5UCAAA9 to the complementary nucleotides 5AGUUU9 while maintaining the linker length, to generate the *P2 lin(5–9)* mutation (orange and light green in Figure 2A), strongly impaired the ability to activate PKR. In this context, 5UC6 (orange in Figure 2A), which form part of pseudoknot stem P2, are the two critical nucleotides, because mutation of 7AAA9 alone to 7UUU9 failed to impair PKR activation.

In conclusion, throughout its 104-nucleotide sequence, the *TNF* RNA element that activates PKR is most sensitive to mutations, rendering it, like the *IFNG* RNA activator of PKR, a prime target for inflammatory disease upon mutation.

### 2.3. The RNA Activator of PKR and Silencer of PKR Activation within Human Globin Pre-mRNA That Controls mRNA Splicing

Efficient splicing of adult *α-globin* and *β-globin* as well as of fetal *Aγ-globin* pre-mRNA species requires activation of PKR and eIF2α phosphorylation within the nucleus [17]. Splicing of *β-globin* mRNA in intact cells is abrogated by 2-aminopurine or by co-expression of dominant-negative mutant PKR [17]. Like *IFN-γ* mRNA, *β-globin* pre-mRNA activates PKR in vitro [17]. Once activated, PKR must phosphorylate its eIF-2α substrate to promote assembly of early spliceosomes and to enable efficient splicing of *β-globin* and *α-globin* mRNA, a key step towards erythropoiesis. Expression of non-phosphorylatable mutant eIF2αS51A inhibits splicing in cells, as do antibodies against phospho-eIF2α in vitro [17]. Once *β-globin* intron 1 is excised, sequence *S1c* near the 5′-terminus of exon 2 (red in Figure 3) induces a structural rearrangement within the RNA that silences the ability of spliced *β-globin* mRNA to activate PKR. As a result, activation of PKR is highly transient, serving only to enhance splicing while avoiding inhibition of β-globin protein synthesis [17,25].

The 5-terminal 124 nucleotides in *β-globin* exon 1, comprising the 50-nucleotide 5′-UTR and the first 74 nucleotides of the open reading frame, constitute the RNA activator of PKR (Figure 3A). Deletion of nucleotides 119AGUUGG124 (purple in Figure 3A) led to almost complete loss of the ability to activate PKR; deletion of nucleotides 1ACAU4 from the 5′-end (light purple in Figure 3A) strongly reduced PKR activation.

Mutation *s1a* of 47CACCA51 (yellow in Figure 3A) to the nucleotides in the opposite strand reduced the ability to activate PKR, whereas mutation *s1b* of 110CGUGG114 (grey in Figure 3A) to the nucleotides in the opposite strand completely abrogated the ability to activate PKR. Combining both mutations, in *s1ab*, created a new 5-base pair helix and fully restored the activation of PKR, showing that this part of the *β-globin* RNA activator of PKR is not orientation-sensitive. Hence, the *S1a-S1b* helix in Figure 3A constitutes an essential core of the *β-globin* RNA activator of PKR, rendering it a sensitive target for mutational disease.

During the differentiation of erythroid cells, there is a dramatic increase in the cellular globin content, which increases from less than 0.1% of total protein in the proerythroblast to 95% in the reticulocyte, reflecting a massive translation of globin mRNA [26]. Erythropoiesis would be compromised severely if the mature spliced *β-globin* mRNA should retain the ability to activate PKR and thereby inhibit its own translation severely, as exemplified by *IFNG* mRNA [15] reviewed above. This defines a need for a mechanism that silences the ability of mature *β-globin* mRNA to activate PKR once splicing has taken place.

Once intron 1 is excised, and exon 2 is joined to exon 1, the 5-nucleotide *S1c* sequence U148-G152 (Figure 3A) located within the first ten nucleotides of exon 2 in spliced *β-globin* mRNA displaces strand *S1b* from *S1a*, which results in a major structural rearrangement (Figure 3B) that silences the ability to activate PKR, thereby allowing unimpeded translation of *β-globin* mRNA in the cytoplasm, essential for survival. Mutation of the 5′-CACCA-3′ motif in *S1a* (yellow in Figure 3B) to 5′-GUGGU-3′ restored the ability of the spliced *β-globin* mRNA to activate PKR, as did mutation of the *S1c* motif to 5′-ACCAC-3′. The double mutation, which restores base pairing between *S1a* and *S1c*, restored silencing of PKR activation even though the orientation of the two complementary strands was inverted [17]. Accordingly, mutations within each of *S1a* or *S1c* are deleterious, rendering them highly sensitive targets for disease.

*α-Globin* pre-mRNA activates PKR as strongly as does *β-globin* mRNA. Splicing of *α-globin* pre-mRNA depends likewise on the activation of PKR and on eIF2α phosphorylation. However, within *α-globin* pre-mRNA, the locations of PKR activator and silencer are reversed, the silencer mapping into exon 1 and the PKR activator into exon 2 [17].

The two fetal *Aγ*- and *Gγ*-*globin* genes are related more closely to the adult *β-globin* gene. Except for a single nucleotide, at position 25 in the 5′-UTR, *Gγ*- and *Aγ*-*globin* gene sequences are identical through exon 2 [27]. Activation of PKR and phosphorylation of eIF2α also regulate excision of *Aγ*-*globin* intron 1 [17]. The *Aγ*-*globin* RNA activator of PKR and the silencer element reside within exon 1 and the first 172 nucleotides of exon 2 [17], but their precise locations have not yet been determined.

Hence, the *β-globin* gene, and by inference the *α-globin* and fetal *γ-globin* genes, each represent RNA targets for disease resulting from mutations that affect either their ability to activate PKR or their ability to silence PKR activation after mRNA splicing has taken place, which is critical for allowing the maximal rate of translation of the mature globin mRNA products, essential for hemoglobin production, for breathing and thus for the survival of humans.

The human gene mutation database for the *β-globin* gene (HBB in www.hgmd.org) contains a number of β-thalassemia mutations that map into the RNA activator of PKR or into the silencer element [25]. These β-thalassemia mutations include 110C and 117G within the *S1b* strand at the core of the PKR activator structure and U148 in the spliced mRNA that falls within the silencer sequence (Figure 3) [25]. However, it is not known as yet whether any of these numerous β-thalassemia mutations affect the ability of *β-globin* pre-mRNA to activate PKR essential for maximal splicing, or to silence PKR activation after mRNA splicing has taken place, which is critical for the translation efficiency of *β-globin* mRNA.

## 3. Materials and Methods

### 3.1. Generation of Mutations in Intragenic RNA Activators of PKR

Generation of mutations within the human gene encoding *IFNG* RNA was described previously [15,16]. Generation of mutations within the human gene encoding *TNF* RNA was described previously [13,14]. Generation of mutations within the human genes encoding *α-globin*, *β-globin,* and *Aγ-globin* RNA was described previously [17]. The human gene mutation database for the *β-globin* gene (HBB in www.hgmd.org) was analyzed previously [25].

### 3.2. Analysis of Mutations in Intragenic RNA Activators of PKR That Affect mRNA Splicing and Translation

The functional impact of mutations generated within the human *IFNG*, *TNF,* and *globin* genes that were created in order to delineate the intragenic RNA activators of PKR [13,14,15,16,17] is analyzed in detail in Section 2 above and represented by color in Figure 1, Figure 2 and Figure 3.

### 3.3. Generation of Three-Dimensional Structure Model of the TNF Intragenic RNA Activator of PKR

The strategy, software, and geometric refinement method used to generate the three-dimensional structure model of the human *TNF* RNA activator of PKR (Figure 2B) have been detailed previously [14].

## 4. Conclusions

The use of PKR activation by intragenic RNA elements that are essential for the regulated expression and function of human genes vital to survival opens up a novel area for RNA-mediated disease. As illustrated by the *IFNG*, *TNF,* and *β-globin* genes, mutations of nucleotides composing these RNA elements that were created in the laboratory in order to deconvolute their structure and function have also revealed novel potential targets for disease. Given the great magnitude of regulation of splicing or translation imparted by intragenic RNA activators of PKR, such mutations render the activation of PKR not only essential to gene expression but also a source of disease. Future research will show to what extent *cis*-acting intragenic RNA activators of PKR are used to regulate gene expression within the human genome. Such human RNA elements offer not only novel opportunities for disease but also a challenge for molecular biologists to detect and define them. The discovery that even a virus, human immunodeficiency virus (HIV) in the first example, co-opts a cellular antiviral mechanism, activation of PKR by RNA encoded within the viral genome, to enable splicing of its own *rev/tat* mRNA [28] supports the concept that PKR-mediated gene regulation via intragenic RNA activators is likely to be far more widespread.

## Figures and Tables

**Figure 1 ijms-25-01323-f001:**
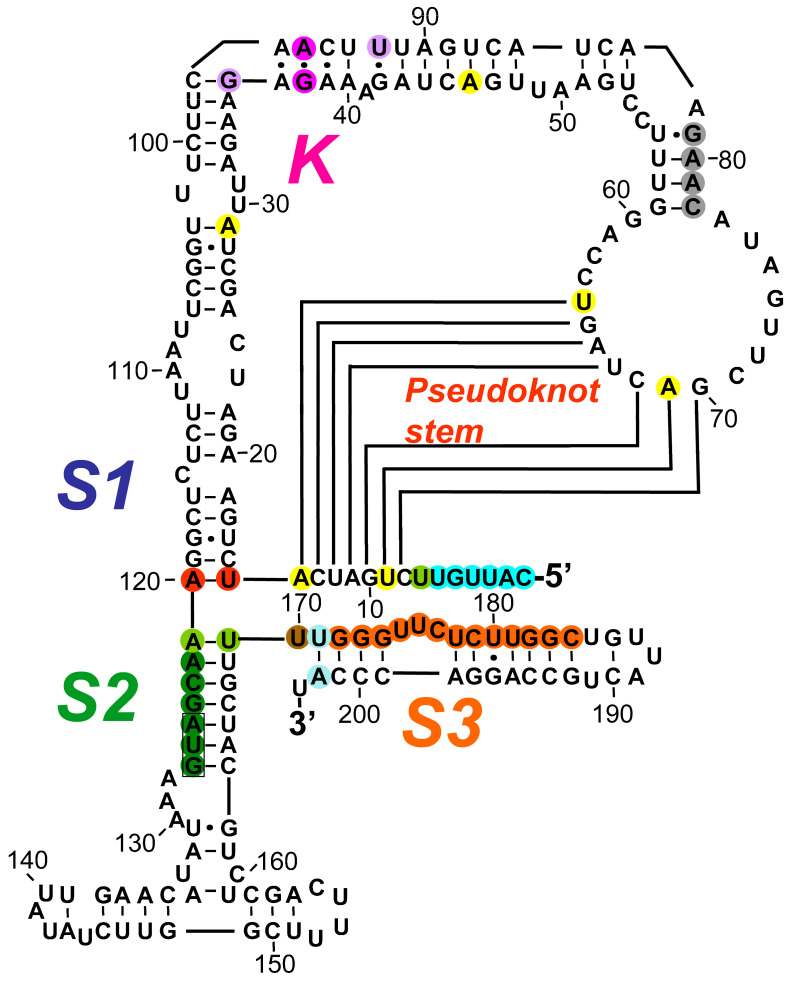
The RNA activator of PKR in human *IFNG* mRNA. The *cis*-acting 203-nucleotide element within human *IFNG* mRNA that activates PKR, to strongly attenuate its own translation, is composed of the entire 5′-UTR and 78 nucleotides of the open reading frame. Secondary structure was determined by T1, U2, and V1 nuclease sensitivity mapping, in-line structure probing in the absence or presence of recombinant PKR and by directed mutagenesis. The pseudoknot structure of the RNA activator of PKR contains the pseudoknot stem, three helices *S1*, *S2,* and *S3*, and kink-turn *K*. The AUG start codon is boxed. Mutations shown to affect its ability to activate PKR are indicated in color [15,16].

**Figure 2 ijms-25-01323-f002:**
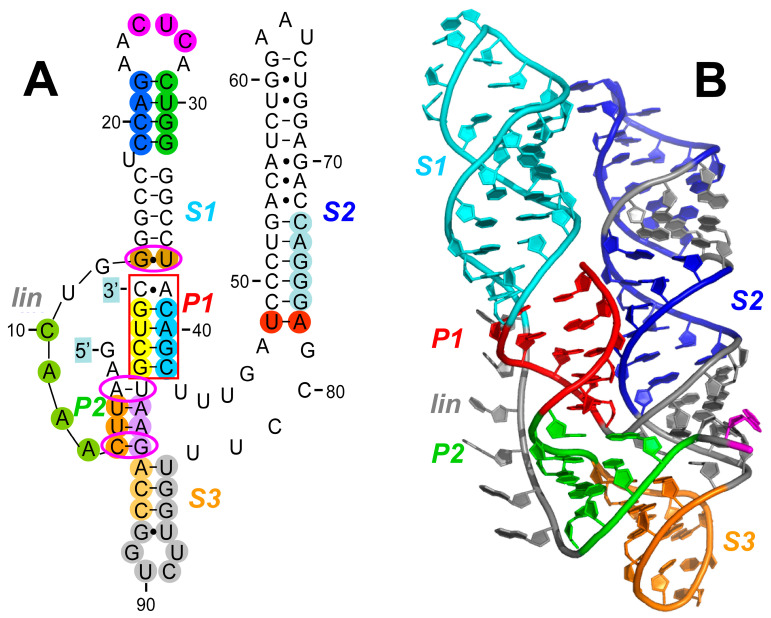
The RNA activator of PKR in human *TNF* pre-mRNA. (**A**) The 104-nucleotide element within human *TNF* pre-mRNA that strongly activates PKR, thereby potently enhancing splicing into mature *TNF* mRNA, is located within the 3′-UTR [13]. Secondary structure was determined by T1, U2, and V1 nuclease sensitivity mapping, in-line structure probing in the absence or presence of recombinant PKR and by directed mutagenesis. The pseudoknot structure of the RNA activator of PKR contains pseudoknot stems *P1* and *P2*, three helices *S1*, *S2,* and *S3*, and linker *lin*. Mutations shown to affect its ability to activate PKR are indicated in color [14]. (**B**) In the folded RNA structure [14], pseudoknot stems *P1* and *P2*, three helices *S1*, *S2,* and *S3*, and linker *lin* are shown in color.

**Figure 3 ijms-25-01323-f003:**
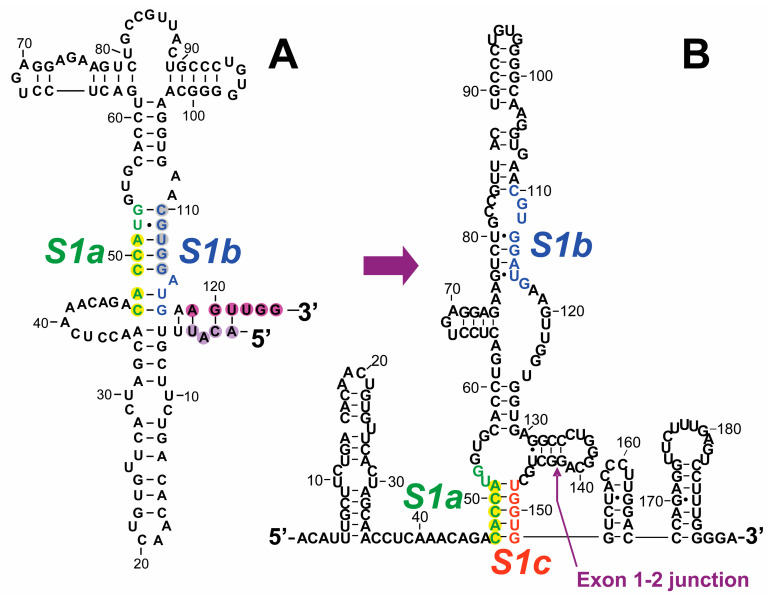
The RNA activator of PKR within human *β-globin* pre-mRNA. (**A**) The 124-nucleotide element within human *β-globin* pre-mRNA that strongly activates PKR, thereby greatly enhancing splicing into mature mRNA, is composed of the entire 5′-UTR and first 74 nucleotides of the open reading frame within exon 1. Secondary structure was determined by in-line structure probing in the absence or presence of recombinant PKR and by directed mutagenesis. The core of the RNA activator of PKR contains the *S1a:S1b* helix. Mutations shown to affect ability to activate PKR are indicated in color [17]. (**B**) The silencer of PKR activation is composed of 5 nucleotides of *S1c* located within the first 10 nucleotides of exon 2. Displacement of strand *S1b* from strand *S1a* by the *S1c* silencer element upon excision of intron 1. Mutation of nucleotides in red color (*S1c*) abrogates silencing of the ability to activate PKR after intron 1 is spliced out [17].

## Data Availability

All data are presented and referenced in the manuscript.

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
