# Peer review of "RNA Activators of Stress Kinase PKR within Human Genes That Control Splicing or Translation Create Novel Targets for Hereditary Diseases"

_ijms, 2024, doi:10.3390/ijms25021323_

Round 1

Reviewer 1 Report

Comments and Suggestions for Authors

The presented manuscript “RNA activators of Stress Kinase PKR within Human Genes that Control Splicing or Translation Create novel Targets for Hereditary Diseases” by Raymond Kaempfer summarized a series of mutation in human RNA-activated protein kinase R (PKR) and characterized its RNA structures and functions, which may provide insights into novel potential RNA targets for human disease. But a few of minor concerns should be addressed before it can be considered for publication.

1. Line 90: Figure 1 I suggest that the author clearly indicate the mutations with different colors in the figure legend. i.e. “a4 mutation (U9G, A14C, A29G, A45U) in yellow color”

2. Line 92 - 102: Although the author cited the references [15, 16] in the figure legend, it is suggested that the relevant literatures should be cited in the text.

3. Line 340- 353: This hypothesis presented the analysis based on the previous mutations published before. It is not necessary to describe the methods and materials.

Author Response

See the PDF.

Reviewer 2 Report

Comments and Suggestions for Authors

In this Hypothesis/Review paper, Dr. Kaempfer summarized some of the key findings over the past two decades from his lab around human intragenic RNA elements that activate PKR and therefore regulate pre-mRNA splicing and/or mRNA translation. This is a great example to show non-coding regions of human mRNAs can have very important regulatory functions and, as the author proposed, could be critical elements involved in human diseases. This reviewer believes this paper represents a very interesting topic and should have a broad audience. I just have several comments which may improve the manuscript:

1. Although review paper(s) about these intragenic RNA elements were already published from the same lab, this current manuscript focuses on RNA structure and critical nucleotides/regions in the intragenic RNAs for PKR activation. Related to this, it would be helpful if the author clearly states how the secondary structures of all three RNAs were obtained. It appears these are secondary structure models build based on chemical probing methods?

2. How is the three-dimensional structure in Figure 2B obtained? Is this a structural model folded by some software? If this is the case, why the author did not get similar models for the other two RNAs?

3. Section 3, Materials and Methods. For this review paper, it appears all the experiments/results were conducted and reported in the original research paper. Is this section required by the journal? If not, this section seems better to be removed.

4. The secondary structure in Figure 3A is a little misleading. The sequence is after removing the intron-1 and this sequence will never fold into this secondary structure. This PKR activator structure exists in the pre-mRNA before removing intron-1. To match the figure legend, the sequence shown here should be the pre-mRNA with intron-1 rather than the same sequence as in 3B which is the matured mRNA. I was confused at the beginning when I try to examine the sequence and structure of Figure 3A.

Minor points:

- Line 99, what are U64 and A69 mutated to?

-Line 112-113, "By contrast, the ability to activate PKR remained intact when all of the nucleotides in helix S1 (U15-A29:U103-A120) except the A120:U15 pair were replaced by other nucleotides.". Please specify what are "other nucleotides". Are these nucleotides still maintain a helical conformation for this region?

-Line 121-122. In s2ab, the base pair is U121:A169. Replace this by the original A121:U169 restores PKR activation capability. Thus, the description should be something like: replacement of the helical junction base pair  U121:A169 by the original A121:U169 within s2ab led to full recovery...

-Line 138, "Hence, A120 and A202 are mutation targets for inflammatory disease". Are there references for this? If not, this should be "potential" targets for inflammatory disease.

Author Response

See the PDF.

Reviewer 3 Report

Comments and Suggestions for Authors

The article briefly discusses the author's previous work demonstrating how RNA regulatory structures embedded within the genes encoding certain cytokines (IFN-ƴ, TNF) and hemoglobin subunits can activate the intracellular kinase PKR, which in turn mediates vital processes like modulating mRNA splicing or translation efficiency. By generating mutations within these intragenic RNA regulator elements and assessing impacts on PKR activation in those past studies, the author suggests such mutations could potentially disrupt normal gene regulation and cause inflammatory or blood diseases. However, this current article does not seem to present much in the way of new findings, instead extracting a few summarized examples from the author's prior publications as illustrative cases to reinforce the notion that mutation of such RNA regulatory elements could have pathological implications.

It seems that this manuscript is summarizing findings from several previous publications by the author rather than presenting much in the way of new research or novel insights. Specifically, it appears to be extracting key bits of information from 5 of the author's prior papers (references 13-17) and consolidating this into a brief overview article. However, without substantially extending upon the earlier work or offering further interpretation, I would agree that there may be limitations in terms of scientific significance or advancement here. The novelty or broader significance of these synthetically introduced mutations remains unclear without further examination or examples beyond what has already been published in the author's previous work.

Author Response

See the PDF.
